# Antibiotic Stewardship in Disaster Situations: Lessons Learned in Lebanon

**DOI:** 10.3390/antibiotics11050560

**Published:** 2022-04-22

**Authors:** Anita Shallal, Chloe Lahoud, Marcus Zervos, Madonna Matar

**Affiliations:** 1Division of Infectious Diseases, Henry Ford Hospital, Detroit, MI 48202, USA; ashalla2@hfhs.org (A.S.); mzervos1@hfhs.org (M.Z.); 2Division of Infectious Diseases, Brigham & Women’s Hospital, Harvard Medical School, Boston, MA 02115, USA; clahoud@bwh.harvard.edu; 3Office of Global Affairs, Wayne State University School of Medicine, Detroit, MI 48202, USA; 4Division of Infectious Diseases, Notre Dame des Secours University Hospital, Byblos 1401, Lebanon; 5School of Medicine and Medical Sciences, Holy Spirit University of Kaslik, Byblos 1401, Lebanon

**Keywords:** antimicrobial stewardship, disaster preparedness, antimicrobial resistance, global health

## Abstract

A post-prescription review and feedback program was implemented as an antimicrobial stewardship intervention in Lebanon as the country grappled with complete economic collapse, the COVID-19 pandemic, and a large blast in Beirut. We describe the implications of antimicrobial use in disaster preparedness and crisis situations, the sequelae related to increasing antimicrobial resistance, and our lessons learned in Lebanon. We explore opportunities and potential solutions for future disaster preparedness.

Antimicrobial resistance (AMR) is a serious threat to global health, costing health care systems over USD 20 billion [1] and is anticipated to result in the death of 10 million people per year by 2050 [2]. In 2019 alone, there were an estimated 1.27 million deaths attributable to AMR [3]. As the global crisis of AMR continues to spread, there is an urgent need for antimicrobial stewardship (AMS) and AMS programs (ASPs). The global burden of infectious diseases is disproportionate and will create a devastating impact across low- and middle-income countries (LMICs) [4]. Within LMICs, the threat of antibiotic overuse has been affected by issues of access, oversight of prescription, and lack of diagnostic tools to support de-escalation. Antibiotics have effects far beyond the patient and individual, as antibiotics contaminate meat and poultry for human consumption, and seep into water sources and agriculture [4]. In the Middle East and North Africa (MENA) region, the social, political, and economic environments vary widely from one another, and major gaps exist with regard to healthcare expenditure and budgeting. For most MENA countries, antimicrobials are available over the counter, and prescription regulation in the community setting is nearly absent [5]. A lack of adequate facilities and comprehensive surveillance programs, coupled with conflict and civil unrest, has led to a six-fold increase in resistance rates since AMR surveillance data began in this region in 2017 [5]. In Syria, for example, a surge of AMR was noted after the onset of several political and economic issues [5]. This was coupled with an alarming increase in the burden of all infectious diseases among Syrians, including outbreaks of measles, poliomyelitis, hepatitis A, bacterial meningitis, and typhoid fever [6]. Among Libyan war casualties, the prevalence of multidrug resistant organisms (MDROs) was 59%, and extended spectrum B-lactamase-producing *Enterobacterales* were the most common [7]. In the MENA region, there is a striking incompatibility between the extent of the problem, and the actions being taken [8].

Outpatient antibiotic treatment and stewardship is equally as important as inpatient. In the outpatient and community setting, self-medication due to the availability of over-the-counter antibiotics has been a real threat to stewardship as well [9,10]. Outpatient management of antimicrobial prescribing is non-controllable in many regions in the Middle East. In Lebanon, most outpatient antibiotic prescriptions are broad spectrum, including quinolones and amoxicillin-clavulanate [11]. Patients who access these antibiotics without a medical prescription do so through auto-medication and dispension of antibiotics by pharmacists, and in one study, just 30% of cases had a true indication for antibiotics [11]. Prior to the pandemic, the number of MDRO infections in the community in Lebanon was readily becoming a concern. In one study, the prevalence of MDR *Enterobacterales* fecal carriage among elderly nursing home residents was 76.5%, a result which was relatively high when compared to similar studies conducted worldwide [12] (for example; 70.3% in Italy [13], 41.3% in Japan [14], and 14.7% in Australia [15]. Among the pediatric community, one study revealed that 34.5% of healthy pediatric patients were found to carry extended-spectrum beta-lactamase producing *Enterobacterales* [16]. These rates of resistant carriage are considerably elevated and uptrending; as shown in one Lebanese study, rates of carbapenem resistant *Klebsiella pneumoniae* had doubled in 2017 [17].

As part of initiatives to reduce AMR and implement ASPs, a post-prescription review and feedback program (PPRF) was built in a tertiary health care facility in Lebanon where baseline, intervention, and post-intervention data were collected. Over the course of the 24 months of this program, the country and people of Lebanon underwent a number of different crises, from the fall and eventual collapse of the Lebanese economy, to the waves of hospitalizations and severe disease from COVID-19, a shortage of healthcare providers, and the devastation of the Beirut blast, which was one of the most powerful explosions in the world after Hiroshima and Nagasaki [18]. The concurrent timeline with the PPRF project provided a unique opportunity to observe how prescription antimicrobials were affected by these crises. The type of antimicrobial agent clearly differed significantly between phases. Through the intervention period, there was a reduction in use of carbapenems due to stewardship efforts. However, in the post-intervention period, which occurred during a time of devastating shortages related to the COVID-19 pandemic and economic collapse, there was noted to be an increase in cephalosporin and carbapenem use when compared to the baseline period, and an increase in aminoglycoside use in the post-intervention period when compared to the intervention period. This was a direct result of significant shortages in the pharmacy, leaving providers to only prescribe what was readily available. This economic turmoil greatly affected AMS in Lebanon, as evidenced in our study. This, coupled with a number of political issues, has led to changes in stocks of antimicrobials from brand to the lowest priced generic product, which may not necessarily be approved for use by governing bodies. The source of these pharmaceuticals may not always undergo the same rigorous quality assurance testing to validate pharmacokinetics and pharmacodynamics, resulting in the possibility of inadequate tissue penetration of multidrug resistant organisms. Following the chaotic prescription of antimicrobials induced by the economic collapse, there has been a delay in observing the subsequent resulting rates of carbapenem resistant organisms (CROs), which this group intends to observe and report. 

In addition to issues with the stockpile of antimicrobials and other therapeutics, the hospitals faced critical shortages of diagnostic tools, including rapid tests such as hemo-glucometers. For several months, some facilities lacked intravenous contrast for CT scanning. From an antimicrobial stewardship standpoint, blood culture bottles, bacterial agar and culture plates, and antibiotic micro-dilution tests are also currently facing critical shortages. These shortages have resulted in less readily available microbiologic data for patient care—thus, resulting in more empiric antimicrobial therapy. Today, for example, a patient who presents with fever and chills is likely to receive broad spectrum antimicrobials without de-escalation of antibiotics due to the absence of necessary microbiologic data. In such challenging situations, physicians aim to reach the correct diagnosis with minimal tests and imaging, relying on clinical data as is frequently done in other low- and middle-income countries [19]. These events all begged the question—what is the role, if any, of antimicrobial stewardship teams in disaster preparedness and crisis situations?

In the event of a disaster, large quantities of pharmaceutical and medical supplies may be required with little to no warning. The WHO expert advisory panel recommends 182 antimicrobial medicines on its “Essential Medicines List” for preparedness. However, pharmaceutical surge capacity is a well-established gap in disaster preparedness [20]. The pandemic lockdowns notably disrupted production, supply, and distribution of medications and changed the stockpile of drugs around the world. When it comes to preparedness, many institutions do not have a dedicated reserve for disaster events. In one report from Maryland, US, 92% of surveyed hospitals had assessed pharmaceutical inventory with respect to biologic agents, and only 64% had reported an additional dedicated reserve supply for biological events, 67% for chemical events, and 50% for radiological events [21]. Hospitals generally remain underprepared for chemical, biological, radiological, nuclear, or explosive attack—and limited supplies of antibiotics for treatment exist [20]. Mass gatherings, though less frequent in the post-COVID era, are also public health issues that command disaster preparedness for hosting cities and countries. These events have the potential to transmit many communicable diseases, frequently compounded by water/sanitation issues. Thermal disorders, stampedes, trauma/crush injuries, and terrorist incidents are potential crises that could arise [22].

Hospital and healthcare facilities are only one component of a coordinated response for disaster management, but they do represent a critical link in the system [23]. The Syrian conflict, which started in 2011, has produced one of the largest refugee crises in modern times, and Lebanon is a home for more Syrian refugees per capita than any other country in the world [24]. These migrant populations may be uninsured or experiencing poverty, and are less likely to seek care unless severely ill, increasing the acuity of patients presenting to hospitals in disaster situations [23]. This can be intensified by the limited number of trained medical professionals due to emigration [25], and some testing may be outsourced due to unavailability of lab resources, or due to the exodus of skilled laboratory staff [5]. During the Syrian civil war, over 60% of hospitals and clinics were destroyed, approximately 700 medical staff were killed, and thousands more fled to neighboring countries [26]. In times of crises, this exodus of highly skilled workers directly impacts the ability to deliver emergency responses. A failure to train hospital members broadly when planning for disaster can be further exacerbated by these shortages of healthcare personnel. Staffing thus becomes a key essential element for expanding surge capacity and preparedness, in addition to infrastructure and supplies [23]. 

During the initial surge of the COVID-19 pandemic, many countries violated the International Health Regulations of the WHO by barring exports of medical supplies in order to prevent the virus from spreading within their own borders [27]. Hindsight has clearly shown this self-over-other strategy has been an ineffective way to address disaster preparedness. An innovative strategy for supply chain management is needed to protect the supply of essential medicines that includes antimicrobials [28]. One potential solution that has been suggested has been to work with local community organizations, including schools and religious buildings to identify potential sites for patient treatment, as well as an area for storage of supplies and equipment [23]. In addition, mutual aid agreements between neighboring hospitals, communities, and countries can help estimate collective capacities, assisting in sharing personnel, equipment, and supplies according to need.

A number of potential solutions are possible when planned in advance. These strategies must be considered now, before the next crisis inevitably occurs. Taking into account cost and political climate will help to achieve a level of “meaningful preparedness” [23]. AMS should be a priority of all health ministries, best strategically implemented in a healthy nation in collaboration with healthcare institutions and public health bodies. The stewardship team certainly has an important role to play in crisis preparedness—the aggressive de-escalation of a single dose of unnecessary antimicrobial can indeed help preserve the medication in situations of limited stock. Building up ASPs across hospital sectors and community pharmacies can help reduce inappropriate antimicrobial use, not only for reducing AMR, but to improve surge capacity and preparedness [28]. The role of the pharmacist within the AMS team is especially important, as they are experts in medication procurement, storage, compounding, and dispensing [29], and thus can develop alternate treatment plans while awaiting stockpiles during a surge. Furthermore, in the context of limited physician access, patients often seek the services of community pharmacists directly, which was shown to be an important opportunity to increase public awareness of AMR in a study in Italy [30]. Investing in adequate staffing to expand hospital surge capacity as needed, ensuring adequate AMS training for personnel, and securing adequate supplies of antimicrobials will be key to minimizing harm in future disaster situations, both in Lebanon and globally [27].

It has been said that the resiliency of a healthcare system is most visible, and most tested, in situations where there is substantial stress and pressure [31]. The world needs to re-examine its capacity for a global response to disaster situations and implement these necessary changes now. We must adopt a unifying global target and need higher income countries to provide logistical, technical, and financial assistance to their neighbors. In the MENA region in particular, there must be legislative changes in the provision of over-the-counter antibiotics in order to reduce unnecessary use in the community setting. Furthermore, there needs to be an emphasis on improving preparedness within public health sectors and organizations with emphasis on training to deal with future pandemics, particularly given the clear evidence for increasing AMR worldwide.

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
