# Peer review of "Antibiotic Stewardship in Disaster Situations: Lessons Learned in Lebanon"

_antibiotics, 2022, doi:10.3390/antibiotics11050560_

Round 1

Reviewer 1 Report

It should be clarified in page 2 line 6 the "theoretical risk for increased promotion of AMR, through use of low levels of drugs or below MICs at the site of infection".

Page 3 line 3  "The role of the pharmacist within the AMS team is especially important..." it should be improved this point explaining how pharmacists could be determinant to reduce AMR in different countries with comparison among them. You should cite some studies to better explain this: (Napolitano F, et al. 2019. The Knowledge, Attitudes, and Practices of Community Pharmacists in their Approach to Antibiotic Use: A Nationwide Survey in Italy.)

Author Response

It should be clarified in page 2 line 6 the "theoretical risk for increased promotion of AMR, through use of low levels of drugs or below MICs at the site of infection". Removed.

Page 3 line 3  "The role of the pharmacist within the AMS team is especially important..." it should be improved this point explaining how pharmacists could be determinant to reduce AMR in different countries with comparison among them. You should cite some studies to better explain this: (Napolitano F, et al. 2019. The Knowledge, Attitudes, and Practices of Community Pharmacists in their Approach to Antibiotic Use: A Nationwide Survey in Italy. Added and included two sentences on this.

Reviewer 2 Report

This is a well-written manuscript with important information on the topic of antibiotic stewardship in disaster situations. Here are some suggestions:

  1. In the last paragraph, the authors make suggestions regarding future actions to implement antimicrobial stewardship practices. Since in some of the countries mentioned (MENA) there is a legislation gap that allows, for example, the provision of antibiotics over-the-counter without medical prescription, it would be important to also suggest legislative changes that could make more strict the provision of antibiotics, in order to reduce unnecessary use in the community setting
  2. In the context of disaster medicine, more examples of AMR and potential for stewardship could be used from MENA countries. For example, there are studies showing a high prevalence of MDR in Libyan war casualties (doi: 10.1089/mdr.2017.0141, 10.1111/1469-0691.12135) or in Syria as well (doi: 10.2217/fmb-2021-0040)

Author Response

This is a well-written manuscript with important information on the topic of antibiotic stewardship in disaster situations. Thank you. Here are some suggestions:

  1. In the last paragraph, the authors make suggestions regarding future actions to implement antimicrobial stewardship practices. Since in some of the countries mentioned (MENA) there is a legislation gap that allows, for example, the provision of antibiotics over-the-counter without medical prescription, it would be important to also suggest legislative changes that could make more strict the provision of antibiotics, in order to reduce unnecessary use in the community setting. Added.
  2. In the context of disaster medicine, more examples of AMR and potential for stewardship could be used from MENA countries. For example, there are studies showing a high prevalence of MDR in Libyan war casualties (doi: 10.1089/mdr.2017.0141, 10.1111/1469-0691.12135) or in Syria as well (doi: 10.2217/fmb-2021-0040) Added.